# Overexposure to Bisphenol A and Its Chlorinated Derivatives of Patients with End-Stage Renal Disease during Online Hemodiafiltration

**DOI:** 10.3390/biom9090403

**Published:** 2019-08-22

**Authors:** Astrid Bacle, Antoine Dupuis, Mohamed Belmouaz, Marc Bauwens, Guillaume Cambien, Nicolas Venisse, Pascale Pierre-Eugene, Sophie Potin, Virginie Migeot, Sarah Ayraud-Thevenot

**Affiliations:** 1INSERM, University Hospital of Poitiers, University of Poitiers, CIC1402, 86021 Poitiers CEDEX, France; 2Biology-Pharmacy-Public Health Department, University Hospital of Poitiers, 2 rue de la Milétrie, 86021 Poitiers CEDEX, France; 3Faculty of Medicine and Pharmacy, University of Poitiers, 6 rue de la Milétrie, 86000 Poitiers, France; 4Digestiv, Urology, Nephrology, Endocrinology Department, University Hospital of Poitiers, 2 rue de la Milétrie, 86021 Poitiers CEDEX, France; 5Research Institute for Environmental and Occupational Health (IRSET-INSERM UMR 1085), University of Rennes 1, 2 Avenue du Pr Léon Bernard, 35043 Rennes, France; 6Pharmacy Department, University Hospital of Rennes, 2 rue Henri Le Guilloux, 35033 Rennes, France

**Keywords:** online hemodiafiltration, hemodialysis, end-stage renal disease, bisphenol A, chlorinated derivatives of bisphenol A, endocrine disruptors

## Abstract

The health safety conditions governing the practice of online hemodiafiltration (OL-HDF) do not yet incorporate the risks related to the presence of endocrine disruptors such as bisphenol A (BPA). The aim of this study was to assess, for the first time, the exposure to BPA but also to its chlorinated derivatives (ClxBPA) (100 times more estrogenic than BPA) during OL-HDF. We demonstrated that BPA is transmitted by the different medical devices used in OL-HDF: ultrafilters, dialysis concentrate cartridges (and not only dialyzers, as previously described). Moreover, BPA has been found in dialysis water as well as in ultrapure dialysate and replacement fluid due to contamination of water coming from municipal network. Indeed, due to contaminations provided by both ultrafilters and water, high levels of BPA were determined in the infused replacement fluid (1033 ng.L^−1^) from the beginning of the session. Thus, our results demonstrate that dialysis water must be considered as an important exposure source to endocrine disruptors, especially since other micropollutants such as ClxBPA have also been detected in dialysis fluids. While assessment of the impact of this exposure remains to be done, these new findings should be taken into account to assess exposure risks in end-stage renal disease patients.

## 1. Introduction 

Due to the increasing incidence of people suffering from end-stage renal disease (ESRD), more than 1 million patients around the world require renal replacement therapy [1]. Hemodialysis (HD) is the most widely used standard therapy for the replacement of renal function. Conventional HD is mainly based on the diffusive transport of solutes across a high flux membrane removing low- and middle-molecular weight uremic toxins. However, HD is poorly effective as a means of clearing larger molecular weight toxins and ineffective in clearing protein-bound toxins, which are associated with increased cardiovascular mortality [2,3]. Online hemodiafiltration (OL-HDF) has recently been proposed to improve clearance of middle sized toxins (of higher molecular weight solutes) and may improve patient outcomes [4,5,6]. Indeed, OL-HDF combines diffusive and convective solute transport through a high-flux permeable membrane. Fluid is removed from the patient by ultrafiltration (convection volume) and appropriate fluid balance is maintained by infusion of a replacement solution, generated online, into the patient’s blood. OL-HDF mode may differ, depending on the site of replacement fluid infusion. Post-dilution OL-HDF, corresponding to infusion of the replacement fluid downstream of the dialyzer, is the most efficient OL-HDF method regarding solute removal [3]. High convection volumes are required in order to maximize the removal of large solutes, leading to intravenous infusion of large volumes of replacement fluid (about 20 L) to maintain fluid balance.

Due to the large volumes of fluid removed from and added to blood during treatment sessions, patients are exposed to risks beyond those associated with HD. The health and safety conditions governing the practice of OL-HDF are well-defined but do not yet include the risks related to the occurrence of endocrine disruptors such as bisphenol A (BPA).

BPA is a man-made chemical frequently used in the production of epoxy resins, polycarbonate (PC) and polysulfone (PS). This monomer has a wide range of applications such as plastic food packaging, coating beverage cans, and medical devices. BPA is widely considered to have an estrogenic effect and epidemiological studies have suggested that human exposure to BPA may be associated with diabetes, cardiovascular diseases, and negative effects on reproductive systems [7,8]. 

BPA has been found throughout the environment, including drinking water from water treatment plants [9] where a chlorination step is often used to conclude treatment operations. The reactivity of BPA with chlorine disinfectants leads to the formation of chlorinated derivatives (ClxBPA). Monochlorobisphenol A (MCBPA), dichlorobisphenol A (DCBPA), trichlorobisphenol A (TCBPA), and tetrachlorobisphenol A (TTCBPA) have been detected in drinking water [10,11,12]. Several in vitro studies have reported adverse effects of ClxBPA, such as endocrine disruption effects. Estrogenic activity of ClxBPA was higher than BPA, especially for mono-, di-, and tri-chloro-BPA [13,14,15]. Indeed, DCBPA was up to 105 times more potent than BPA as a ligand for estrogen receptor [16]. This estrogenic activity resulted in proliferation of breast cancer cells [17] and uterine endometrium cells [18].

ClxBPA also have adverse effects on metabolism and on adipogenesis. ClxBPA showed 10–100 times higher binding affinity to peroxisome proliferator-activated receptors (PPARγ) than BPA. The highest PPARγ agonist activity is observed with TCBPA and TTCBPA and is associated with the appearance of obesity and diabetes [19]. These properties have been highlighted in a zebra fish model in which ClxBPA induced a lipid accumulation as well as in humans, since urinary MCBPA levels were significantly associated with type II diabetes mellitus [20]. 

Finally, similarly to BPA, ClxBPA may affect the thyroid system. Indeed, in a two-hybrid assay in yeast incorporating the human thyroid hormone receptor α, TCBPA, and TTCBPA exhibited an agonist activity and inhibited the binding of triiodothyronine [21]. 

In 2015, the Scientific Committee on Emerging and Newly Identified Health Risks (SCENHIR) concluded that the risk of adverse effects due to BPA may exist in patients undergoing dialysis treatment since BPA release from polymeric medical disposables used in hemodialysis is significant and BPA accumulation in systemic circulation is amplified by reduced renal clearance [22]. 

Only BPA release from dialyzers has been taken into account in risk assessment of exposure in dialysis. However, we have recently demonstrated that BPA was present in other medical devices as well as in water used during HD, leading to significant exposure of patients suffering from ESRD [23]. Moreover, few data are available concerning BPA exposure during OL-HDF treatment sessions, even though patients can be highly exposed to micropollutants via the replacement fluid infused in blood circulation. Furthermore, exposure to ClxBPA in patients receiving renal replacement therapy has never been assessed. Therefore, the aim of this work was to assess, for the first time, the risk of exposure of ESRD patients not only to BPA, but also to ClxBPA during OL-HDF treatment, taking into account the potential critical points of BPA release and assessing BPA and ClxBPA levels in water and dialysis fluids.

## 2. Materials and Methods

This study was conducted at the University Hospital of Poitiers, France. Water, dialysate, and replacement fluid samples were collected in order to determine BPA and ClxBPA contamination levels.

### 2.1. Water Purification Process

Samples were collected at each treatment step required to provide dialysis water from drinking water supplied by a municipal network (Figure 1). Samples were collected on different days (*n* = 6) in order to take into account the variability of target compound levels.

### 2.2. Production of Ultrapure Dialysate

The different sampling points are shown in Figure 2.

#### 2.2.1. Dialysis Concentrates 

BiCart Select^®^ combi-pak concentrates (Gambro, Meyzieu, France) were assessed. They include two polypropylene cartridges: SelectCart^®^ and BiCart^®^ cartridge containing dry sodium chloride and dry sodium hydrogen carbonate respectively. One liter of BPA-free water (Carlo Erba, Peypin, France) was passed through each cartridge (*n* = 6). BPA concentration was then determined in 250 mL of this solution. In the same way, we determined BPA level in 250 mL of two types (CX265G and CX375G) of the Citrate SelectBag^®^ concentrate solutions, providing K^+^, Ca^2+^, Mg^2+^, and glucose.

#### 2.2.2. Ultrafilters

The ultrafilters (U 8000 S, Gambro, France) were made of Polyamix™ (a blend of Polyarylethersulfone, Polyvinylpyrrolidone, and Polyamide) membrane. We determined (*n* = 6) the concentrations of released BPA by circulating ultrafilters with 400 mL of BPA-free water at a flow rate of 600 mL/min as done in practice.

#### 2.2.3. Dialysis Water and Ultrapure Dialysate

We collected samples at the inlet (ultrapure water) and at the outlet (ultrapure dialysate) of the HDF machine (AK200 Ultra S, Gambro^®^, Meyzieu, France) (Figure 2). In order to assess the variations of BPA concentrations throughout a HDF session, samples were taken at different times of dialysate production (hourly sampling for 4 h). Regarding ClxBPA, ultrapure dialysate samples (250 mL) were collected at the beginning of dialysate production. These experiments were repeated six times.

### 2.3. Replacement Fluid 

Replacement fluid is a sterile non-pyrogenic solution prepared on-line from the ultrapure dialysis fluid by an additional step of ultrafiltration using a sterile ultrafilter (polysulfone membrane and polycarbonate housing) integrated in a sterile line set (LINEA HDF on line, Medica^®^, Italia) (Figure 2). In the same way as for ultrapure dialysate, we have collected replacement fluid samples (250 mL) at t = 0 h and then every hour during 4 h in order to determine BPA. ClxBPA levels were determined at the beginning of replacement fluid production. These experiments were repeated 6 times.

### 2.4. Tubing

The dialyzers were connected to the blood and dialysate compartments by tubing (BL200BD PRE POST^®^, Gambro, Meyzieu, France), made of polyvinyl chloride (PVC). The tubes (*n* = 6) were assessed by circulating 400 mL of BPA-free water during 180 min in order to determine BPA leaching.

### 2.5. Dialyzers

We investigated four different HDF dialyzers commonly used in our institution (Table 1). As done in practice in order to remove manufacturing residues [24], we have rinsed the dialyzers using 2 L of 0.9% sodium chloride solution (Clear flex^®^, Baxter, Jonage, France) in which we had previously verified the absence of BPA.

A dialysis session was then mimicked using a previously described in vitro model [23]. Dialysate and blood compartments were recirculated countercurrently with 400 mL of BPA-free water at 250 mL/min for 180 min. Experiments were all carried out at 37 °C, in order to mimic the conditions of a hemodialysis session. Each dialyzer was assessed three times.

### 2.6. Sample Analysis

To avoid contamination, the samples were collected in glassware calcined at 500 °C for 5 h. Solvents and reagents were free of BPA and ClxBPA. 

BPA (CAS 80-05-7) and internal standard (IS) bisphenol A-d16. (CAS 96210-87-6) were obtained from Sigma-Aldrich Inc. (St. Louis, MI, USA). Chlorinated BPA were custom synthesized by @rtMolecule (Poitiers, France). The purity obtained for these compounds was >98%.

Sample preparation and analytical method used to determine the concentration of BPA and ClxBPA were adapted from a previously developed and fully validated method [9] adding an online solid-phase extraction step to UPLC–MS/MS analysis [25]. The validated limit of quantification (LOQ) was set at 0.8 ng·L^−1^ and 0.08 ng·L^−1^ for BPA and ClxBPA, respectively. 

#### 2.6.1. Sample Preparation 

Each target compound (BPA, CBPA, DCBPA, TCBPA, TTCBPA) and internal standard (IS) (BPA-d16 and DCBPA-d12) was weighted and dissolved in methanol to obtain corresponding 200 mg·L^−1^ stock solutions which were stored frozen (−20 °C) before use.

Working standard solutions were diluted in methanol/water 50/50 (*v*/*v*) to obtain the following final concentrations: 2, 4, 8, 20, and 40 µg·L^−1^ and 0.2, 0.4, 0.8, 2, and 4 µg·L^−1^, for BPA and ClxBPA respectively. Initial solutions of BPA-d_16_ and DCBPA-d_12_ were also diluted in methanol to obtain IS working solution at 20 µg·L^−1^ and 2 µg·L^−1^, respectively.

Before extraction, 12.5 mL of methanol and 100 µL of IS working solution were added to an aliquot of 250 mL of water sample to obtain final concentrations ranging from 0.8 ng·L^−1^ to 16 ng·L^−1^ and from 0.08 ng·L^−1^ to 1.6 ng·L^−1^ for BPA and ClxBPA respectively. Solid-phase extraction (SPE) using Oasis HLB (200 mg/10 mL Waters, Milford, CT, USA) glass cartridges was performed in order to purify and concentrate water samples. SPE cartridges were conditioned and equilibrated with methanol (2 × 3 mL) and water (2 × 3 mL), respectively. After these two steps, water samples were loaded onto the cartridges, which were subsequently washed with a mixture of water and MeOH (80/20, *v*/*v*) and allowed to dry for 15 min before elution with 6 mL of methanol. The extracts were then evaporated to dryness at 30 °C under a gentle nitrogen stream. The residues were dissolved in 250 µL of water/MeOH (80/20, *v*/*v*) and finally 20 µL were injected into the SPE-UPLC-MS/MS system.

#### 2.6.2. Online SPE-UPLC-MS/MS

The online SPE-UPLC-MS/MS system consisted in an isocratic pump (ICS^®^, Toulouse, France) for online SPE and an UPLC-MS/MS system composed of an UPLC system Acquity^®^ H Class (Waters, Milford, CT, USA) coupled to a Xevo^®^ TQ-S triple quadrupole mass spectrometer (Waters, Milford, CT, USA).

An Xbridge^®^ C_18_ column (2.1 × 30 mm, 10-µm-particle diameter, Waters, Milford, CT, USA) was used for the fully automated SPE online procedure.

Chromatographic separation was achieved using an Aquity^®^ CSH C_18_ column (1.7 µm particle size, 2.1 × 100 mm, Waters, Milford, CT, USA) heated at 40 °C and a binary mobile phase (MeOH/water) delivered in the gradient mode at a flow rate of 350 µL.min^−1^. Quantification was obtained by using MRM mode with two *m*/*z* transitions per analyte, one for quantification and one for confirmation, in negative electrospray ionization mode 

#### 2.6.3. Method Validation

The limit of detection (LOD) was defined as three times the standard deviation of the signal of five blank samples of BPA-free water [26].

The limit of quantification (LOQ), defined as the lowest concentration that could be reliably quantified, was validated at 0.8 ng·L^−1^ and 0.08 ng·L^−1^ for BPA and ClxBPA, respectively. Calibration curves, ranging from 0.8 to 12.8 ng·L^−1^ and from 0.08 to 1.28 ng·L^−1^, for BPA and ClxBPA, respectively, were regularly checked during analysis for linearity (r^2^ > 0.99). Accuracy was assessed using quality controls (0.8 and 12.8 ng·L^−1^ and 0.08 and 1.28 ng·L^−1^, for BPA and ClxBPA, respectively) prepared using BPA and ClxBPA-free water, (trueness ranged from 80% to 120%, and precision was ≤20%).

#### 2.6.4. Assigning Values to Non-Detected/No-Quantified Data

In the field of environmental health, variables with analytically non-detected or non-quantified values are commonly encountered. For statistical analysis of these data, deletion (ignoring the observations) or substitution with zero methods provide a biased performance. Therefore, in order to minimize this bias, non-detected and non-quantified data were substituted for a fixed value: LOQ divided by the square root of two (LOQ/√2) [27].

### 2.7. Statistical Analysis

Statistical analyses were performed using the Wilcoxon signed-rank-test. BPA concentrations versus time profiles were analyzed by Friedman test analysis with post hoc analysis as appropriate. For all statistical comparisons, significance was defined as *p* < 0.05. Analyses were conducted with SAS1 (version 9.3; SAS Institute, Cary, NC, USA). 

## 3. Results 

### 3.1. BPA Leaching from Tubing and HDF Dialyzers

No BPA leaching was observed from the tubes connecting the dialyzer to blood and dialysate compartments.

In clinical practice, before HDF sessions, dialyzers are rinsed with a solution of 0.9% sodium chloride. Whatever the dialyzer assessed, large amounts of BPA are quantified in rinsing solution ranging from 111.7 ± 48.4 to 440.4 ± 188.3 ng per dialyzer on average (Table 2). After rinsing, BPA levels in our study were determined in simulating blood and in dialysate compartment. BPA was detected in every dialyzer with mean levels ranging from 0.6 to 38.5 ng per dialyzer in simulating blood compartment and from 0.9 to 16.2 ng per dialyzer in dialysate compartment. No significant difference in BPA levels was found between both compartments.

### 3.2. BPA and ClxBPA Occurrence throughout the Water Purification Process

BPA and ClxBPA were detected in water samples collected at the entrance of the water purification process (sp1) and after most steps of the water treatment process (Table 3). In ClxBPA, DCBPA, and TTCBPA were most frequently detected at concentrations reaching 15.30 ± 13.00 ng·L^−1^ and 1.90 ± 2.30 ng·L^−1^ on average, respectively. MCBPA and TCBPA were detected in water samples, but most often at a level below LOQ. Regarding BPA and ClxBPA concentrations, no statistically significant difference was observed between samples collected before and after each water treatment step. 

### 3.3. BPA and ClxBPA in Ultrapure Dialysate and Replacement Fluid

BPA was quantified in dialysis water, at the inlet of the dialysis machine, at a mean concentration of 4.0 ± 1.6 ng·L^−1^. This value was not significantly different from the value obtained at the outlet of the water purification process (sp7). 

The dialysis machine studied contains two ultrafilters. The first one ensures that dialysis water is ultrapure in order to prevent biofilm development inside the dialysis machine. This ultrapure water flows through a cartridge containing bicarbonates and is then mixed with dialysis concentrates leading to the dialysate. In order to prevent microbiological contamination of dialysate from concentrates, this fluid is purified on a second ultrafilter, leading to ultrapure dialysate (Figure 2). We have assessed BPA releasing from these devices. The amount of BPA reached 103.6 ± 44.4 ng·L^−1^ in ultrafilters while low levels of BPA were quantified in bicarbonate cartridges (1.7 ± 0.5 ng·L^−1^ for BiCart^®^ cartridges and 2.0 ± 0.8 ng·L^−1^ in SelectCart^®^ cartridges). The dialysis concentrates assessed, CX265G and CX375G, released approximately the same amounts of BPA, 0.8 ± 0.4 ng·L^−1^ and 0.9 ± 0.3 ng·L^−1^, respectively.

BPA levels were also measured during production of ultrapure dialysate. BPA was detected at the beginning of the fluid production (t = 0 h) at a mean concentration of 20.7 ± 32.0 ng·L^−1^. This value, obtained at the outlet of the dialysis machine, was significantly higher than the value measured at the inlet of the dialysis machine (*p* = 0.002), having increased by almost a factor of five. No statistically significant variation was observed in BPA levels measured during the 4 h of fluid production (Figure 3a). 

Replacement fluid was prepared from ultrapure dialysate by an additional step of ultrafiltration in the dialysis machine studied. In this fluid, BPA levels reached 1033.0 ± 400.8 ng·L^−1^ at the beginning of its production. Replacement fluid concentration was significantly higher than the BPA concentration measured in ultrapure dialysate, having increased by a factor of almost 50 at t = 0 h (*p* = 0.002). This contamination significantly decreased over time during the production process, reaching 265.0 ± 136.0 ng·L^−1^ after 4 h (*p* = 0.001) (Figure 3b). 

ClxBPA was detected in ultrapure dialysate and in replacement fluid as well as in dialysis water (Table 4). No statistically significant differences were noted between ClxBPA levels assessed in ultrapure dialysate and in replacement fluid. In the ClxBPA detected in fluids, DCBPA was determined at much higher concentrations than MCBPA, TCBPA, and TTCBPA. 

## 4. Discussion

This is the first study demonstrating that patients suffering from ESRD and treated by OL-HDF have a risk of overexposure to both BPA and to ClxBPA provided by intravenous infusion of replacement fluid.

### 4.1. BPA Is Leached from High-Permeability Dialyzers 

Just as has been described for dialyzers used in HD [13,14,28,29], our results demonstrate leaching of BPA from high-permeability dialyzers used during OL-HDF. Before OL-HDF sessions, dialyzers are rinsed in order to remove manufacturing residues. Our results highlight that rinsing the dialyzers is essential before a dialysis treatment session in order to reduce BPA release during OL-HDF sessions. Even though substantial amounts of BPA are then eliminated, BPA remains quantifiable in dialyzers. In accordance with previous studies, polysulfone and polycarbonate dialyzers leach the highest amounts of BPA [13,23,29] and their use is associated with significantly increased serum BPA levels among treated patients [30].

A dialyzer with a polypropylene housing is BPA-free certified by its manufacturer. Surprisingly, it still contains BPA but at a lower level than the other dialyzers assessed. Glues used to assemble the dialyzers during the production process could be the source of this contamination [31]. 

In contrast, no BPA leaching was detected from the tubes connecting the dialyzer to blood and dialysate compartments. This result is consistent with information provided by European PVC manufacturers who no longer use BPA in PVC production [22].

### 4.2. BPA and ClxBPA Are Not Eliminated by Water Purification Process

Dialysate and replacement fluids are produced from purified water obtained from treatment of drinking water coming from municipal water network. BPA and ClxBPA were present at the entrance and at each step of water purification process. These results suggest that none of the water treatment steps are effective with regard to these contaminations of dialysis water. Concerning BPA, it has been demonstrated that this compound can be removed by adsorption on carbon and by thin film composite reverse osmosis membrane [32,33]. Our findings support the hypothesis that elimination of BPA by these treatment steps could be compensated by continuous release of BPA from housing and other materials present on water treatment. If no data are available concerning the efficiency of these processes for removal of ClxBPA from water, our results strongly suggest that the studied purification system is unable to reduce contamination of dialysis water from these micropollutants. Moreover, no long-term stability studies exist for Clx-BPA. However, our own experimental unpublished data suggest long-term stability of Clx-BPA in water. Therefore, BPA and ClxBPA concentrations found in water are not supposed to be influenced by ClxBPA transformation.

### 4.3. Dialysate and Replacement Fluid are Sources of Exposure to BPA and ClxBPA 

Ultrapure dialysate is obtained after ultrafiltration of purified water, followed by mixing with dialysis concentrates and by a second ultrafiltration. At the beginning of its production, dialysate fluid was five times more contaminated by BPA than incoming dialysis water. These results suggest that BPA could be continuously released by the machine’s internal fluid pass and/or dialysis concentrates and at a higher rate by ultrafilters. These ultrafilters are made of polycarbonate housing and a Polyamix™ membrane (a blend of polyarylethersulfone, polyvinylpyrrolidone, and polyamide) just like the Polyflux^®^ dialyzer leaching large amounts of BPA.

In the dialysis system studied, replacement fluid is prepared by ultrafiltration of ultrapure dialysate. At the beginning of its production, replacement fluid was fifty times more contaminated by BPA than ultrapure dialysate. This contamination decreased during replacement fluid production but remained higher than contamination of ultrapure dialysate, suggesting that flushing the ultrafilter integrated in the sterile line provides BPA. The filter is a single-use device also composed of a polycarbonate housing and a polysulfone membrane. Thus, in practice, flushing the sterile replacement fluid with ultrapure dialysate when including an ultrafilter could be an appropriate precaution before beginning a treatment session to reduce BPA exposure by OL-HDF.

In addition to BPA, ClxBPA was detected in water contributing to the water purification process, after each treatment step and, finally, in dialysis water. These results suggest that none of the water treatment steps are as means of reducingClxBPA in dialysis water. The same amounts of ClxBPA were detected in dialysis water, in ultrapure dialysate and in replacement fluid. In the ClxBPA detected in fluids, DCBPA was determined at much higher concentrations than MCBPA, TCBPA, and TTCBPA. Previous in vitro studies have demonstrated that this chlorinated derivative has the strongest estrogenic activity: 38- to 105-fold higher than BPA [15,16]. During OL-HDF, DCBPA, as well as other ClxBPAs, are directly and widely available for systemic exposure due to contamination of infused replacement fluid.

After oral route, BPA is rapidly and fully absorbed in the gastrointestinal tract, widely conjugated in bowel and liver (sulfo and glucuroconjugates) and is then rapidly eliminated, leading to blood half-life in less than 2 h [34,35]. BPA is excreted in urine as BPA glucuronide for the most part, and to a much lesser extent in unchanged form. Therefore, in ESRD patients, decline in kidney function leads to the accumulation of BPA, yielding higher blood levels [28,30]. Moreover, during dialysis, the contact of blood with polymeric medical devices, such as dialyzers made of polycarbonate, polysulfone, or polyester, may expose ESRD patients to BPA [23,28,30,36]. In addition to dialyzers, BPA present in dialysate fluid is a source of exposure [23]. To make matters worse, that removal of BPA from blood using dialysis methods is a challenge given the degree of protein-bound fraction of BPA in plasma [37]. In 2015, the Scientific Committee on Emerging and Newly Identified Health Risks (SCENHIR) concluded that the risk of adverse effects due to BPA may exist in patients undergoing dialysis treatment [22]. Cardiovascular risk [38], blood pressure increase [39], and monocyte cytotoxicity [40] are potentially negative effects of BPA in this population. However, long-term follow-up of hemodialyzed patients is required in order to explore long-term negative impact of BPA.

We have demonstrated in a previous work that BPA exposure could reach an estimated 140 ng/kg b.w./day for HD patients. Concerning OL-HDF, infusion of 20 L of replacement fluid provides additional 7840 ng of BPA during a session, corresponding to an additional dose of 112 ng/kg b.w./day compared to HD. The European Food Safety Authority (EFSA) experts have established an oral temporary tolerable daily intake (t-TDI) of 4 µg/kg b.w./day [41]. However, regarding hemodialysis and especially OL-HDF treatment, BPA is directly available for systemic exposure leading to a 100-fold higher oral equivalent dose of exposure. Last but not least, OL-HDF also leads to higher exposure to ClxBPA, yielded by dialysate (up to 3072 ng per session) and, in addition, by replacement fluid infusion (up to 1340 ng per session). Compared to BPA, no pharmacokinetic and toxicodynamic data are available for ClxBPA either in animals or humans. Hence, the metabolism, tissue distribution, and elimination from the body remains unclear and no-TDI has been established for these compounds. Recent studies affirm that OL-HDF could reduce BPA blood levels in dialyzed patients [42,43]. However, the methodology used raises some questions since the number of patients was low, and some issues regarding the analytical method used. Indeed, enzyme-linked immunosorbent assay (ELISA) was performed to determine BPA concentration while this analytical method has been evidenced as non-specific due to cross-reactivity of the anti-BPA [44]. Further studies using an appropriate biomarker of exposure should be performed in order to clarify the role of the replacement therapy on BPA blood levels.

## 5. Conclusions

This is the first study demonstrating that patients suffering from ESRD and treated by OL-HDF have a risk of overexposure to BPA leaching from different medical devices (ultrafilters, dialysis concentrates, and not only dialyzers). Moreover, we have demonstrated that exposure to BPA by drinking water used for ultrapure dialysate and replacement fluid productions must also be considered, especially since water can expose to other micropollutants such as ClxBPA. Exposure to these compounds is of major concern due to infusion of replacement fluid in blood of patients treated by post-dilution OL-HDF. Indeed, using this replacement therapy, large amounts of BPA and ClxBPA are directly available for systemic exposure. Patients are treated for years at a rate of 3 to 4 sessions per week on average are thus chronically exposed to the BPA transmitted by medical devices as well as the BPA and ClxBPA in water (about 170 L for a session). While assessment of the impact of this exposure remains to be done using relevant biomarkers, our findings should be taken into account firstly for improving dialysis water quality. Moreover, other by-products may be taken into account, in particular these resulting from chloramination process applied during the production of drinking water [45]. Finally, the implementation of terminal treatment compatible with the safety rules of hemodialysis practice should be explored in order to remove organic compounds such as BPA and its derivatives.

## Figures and Tables

**Figure 1 biomolecules-09-00403-f001:**
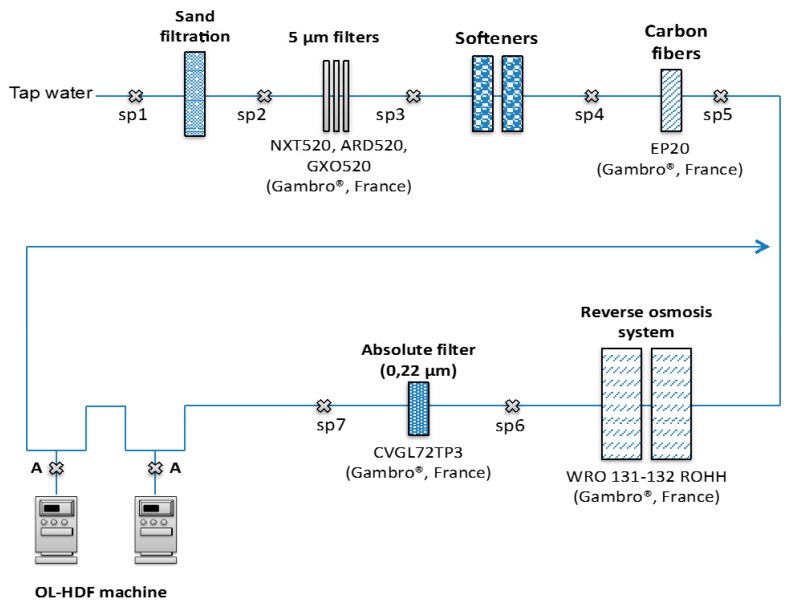
Schematic representation of water purification process and water sample location (
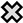
: SP).

**Figure 2 biomolecules-09-00403-f002:**
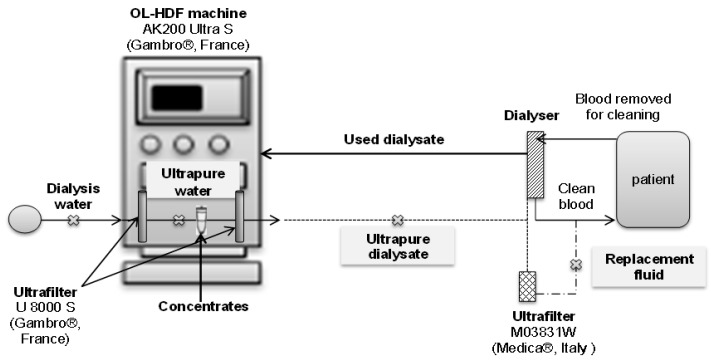
Sample location (
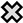
: SP) for ultrapure water, ultrapure dialysate and replacement fluid.

**Figure 3 biomolecules-09-00403-f003:**
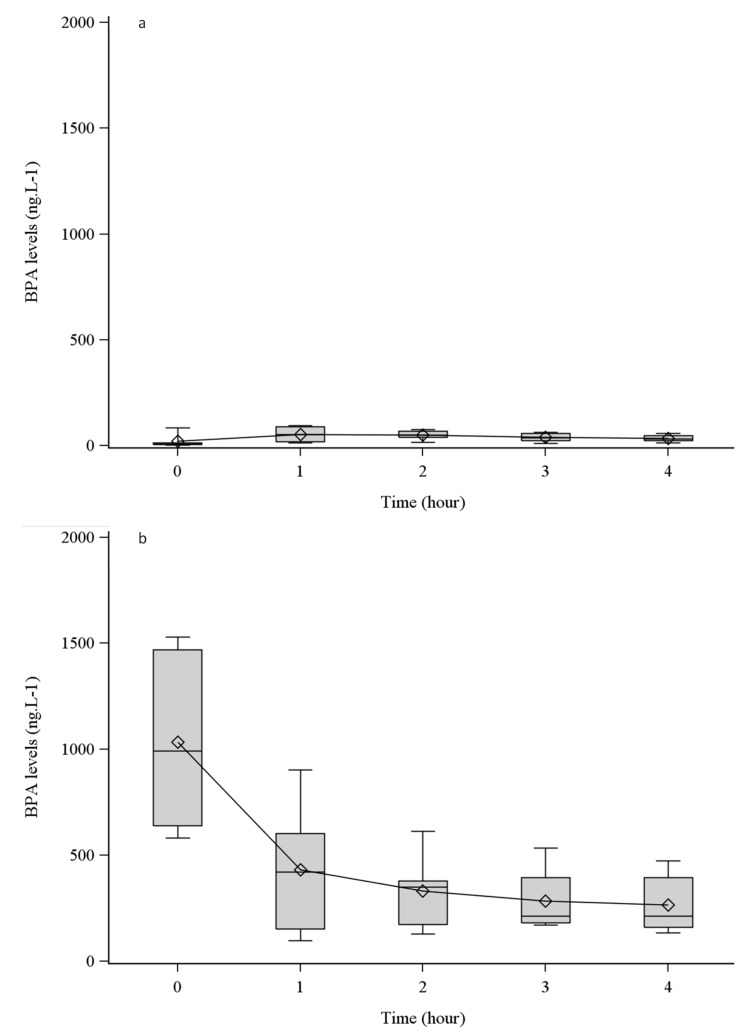
Bisphenol A (BPA) concentrations in ultrapure dialysate (**a**) and in replacement fluid (**b**) versus time profiles. Box plot showing median and interquartile range (IQR, 25th–75th percentile) of BPA ( ◊: mean value).

**Table 1 biomolecules-09-00403-t001:** Hemodiafiltration dialyzers characteristics

Dialyzers	TS-2.1SL	Vie-21A	Elisio-21H	Polyflux 210H
Manufacturers	Toray (Tokyo, Japan)	AsahiKASEI (Tokyo, Japan)	Nipro EUROPE (St.Beauzire, France)	Gambro (Colombes, France)
Housing material	Polycarbonate	Polycarbonate	Polypropylene	Polycarbonate
Fiber material	Polysulfone	Polysulfone	Polyethersulfone	Polyarylethersulfone, Polyvinylpyrrolidone, Polyamide blend
[Inner diameter (µm)	200	185	200	215
Membrane thickness (µm)	40	45	40	50
Effective surface area (m²)	2.1	2.1	2.1	2.1
Potting material	Polyurethane	Polyurethane	Polyurethane	Polyurethane
Sterilization	Gamma-ray irradiation	Gamma-ray irradiation	Dry gamma	Steam

**Table 2 biomolecules-09-00403-t002:** Amount of bisphenol A (BPA) released from the dialyzers used for online hemodiafiltration. Mean ± standard deviation are presented.

Dialyzers	Compartment	BPA (ng/dialyzer)
TS-2.1 SL	Rinsing solution	365.9 ± 266.1
	Simulating blood	0.6 ± 0.5
	Dialysate	0.9 ± 0.1
Polyflux 210H	Rinsing solution	299.0 ± 129.7
	Simulating blood	1.4 ± 0.2
	Dialysate	3.2 ± 3.5
Vie-21A	Rinsing solution	440.4 ± 188.3
	Simulating blood	38.5 ± 41.7
	Dialysate	16.2 ± 5.3
Elisio-21H	Rinsing solution	111.7 ± 48.4
	Simulating blood	3.7 ± 0.5
	Dialysate	3 ± 0.4

**Table 3 biomolecules-09-00403-t003:** Concentrations of bisphenol A and its chlorinated derivatives measured at the inlet and after each treatment step of the water treatment process. Mean ± standard deviation are presented are presented (LOQ = limit of quantification and ND = non detected).

	BPA (ng·L^−1^)	MCBPA (ng·L^−1^)	DCBPA (ng·L^−1^)	TCBPA (ng·L^−1^)	TTCBPA (ng·L^−1^)
Water intake (SP1)	2.8 ± 2.0	0.7 ± 1.3	14.4 ± 14.3	<LOQ	1.9 ± 2.3
Sand filter (SP2)	0.8 ± 0.5	<LOQ	8.9 ± 6.6	<LOQ	0.6 ± 0.4
5 µm filter (SP3)	2.0 ± 2.4	<LOQ	15.3 ± 13.0	<LOQ	1.0 ± 1.2
Softener (SP4)	3.2 ± 4.4	<LOQ	9.7 ± 4.0	<LOQ	0.6 ± 0.6
Carbon filter (SP5)	4.6 ± 7.6	<LOQ	12.9 ± 9.0	ND	0.9 ± 0.8
Reverse osmosis (SP6)	6.1 ± 10.0	<LOQ	14.5 ± 11.5	ND	1.0 ± 1.2
Absolute filter (SP7)	3.1 ± 2.8	<LOQ	8.4 ± 9.6	ND	0.6 ± 0.9

BPA = bisphenol A; MCBPA = monochlorobisphenol A; DCBPA = dichlorobisphenol A; TCBPA = trichlorobisphenol A; TTCBPA = tetrachlorobisphenol A.

**Table 4 biomolecules-09-00403-t004:** Concentration of bisphenol A chlorinated derivatives in ultrapure water and at the beginning of ultrapure dialysate and replacement fluid production. Mean ± standard deviation is presented.

	Ultrapure Water (ng·L^−1^)	Ultrapure Dialysate (ng·L^−1^)	Replacement Fluid (ng·L^−1^)
MCBPA	0.4 ± 0.9	1.6 ± 1.7	3.4 ± 3.2
DCBPA	6.5 ± 1.2	12.8 ± 5.9	42.5 ± 22.1
TCBPA	1.8 ± 1.3	1.3 ± 0.7	1.3 ± 0.6
TTBPA	1.4 ± 0.9	0.7 ± 0.6	0.5 ± 0.4

MCBPA = monochlorobisphenol A; DCBPA = dichlorobisphenol A; TCBPA = trichlorobisphenol A; TTCBPA = tetrachlorobisphenol A.

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
