# Peer review of "Overexposure to Bisphenol A and Its Chlorinated Derivatives of Patients with End-Stage Renal Disease during Online Hemodiafiltration"

_biomolecules, 2019, doi:10.3390/biom9090403_

Round 1

Reviewer 1 Report

I enjoyed reading the study by Bacle and colleagues and believe the issue raised deserves more awareness in the field and therefore think the study is of interest. The results are interesting and important, the introduction very thorough and the methodology mostly well described. However, some points need to be addressed before the manuscript should proceed:

1) Most importanty, the text needs to be revised by either a native speaker or better by a professional English language editing service. There are a number of grammar and syntax issues that should be improved upon. (e.g. abstratct: "provided by the whole medical devices"; line 146 "sodium chloride solution in whom.."; etc)

2) Repeatedly, the authors give mean +/- SD data in the text, while providing median and IQR data in the tables. This makes it challenging to the reader to understand which data the authors are writing about. Please harmonize, I assume data are normally distributed and give mean +/- SD (+n-number) data to more fully inform what the measurements looked like.

3) I believe the order of appearance of the tables has been mixed up. Line 21 to 217: I believe the data in this abstract pertain to table 4 (data on BPA and its derivatives in dialyzer rinsing solution), not to table 2 as is written now.

4) Please provide a concise description of what the ex vivo dialyzer model loooked like. Since dialysate and blood levels were almost equal what did time and flow rates look like? What were BPA levels in the used dialysate to start with?

Author Response

Thank you very much for your valuable and constructive suggestions.

Point 1 : Most importanty, the text needs to be revised by either a native speaker or better by a professional English language editing service. There are a number of grammar and syntax issues that should be improved upon. (e.g. abstratct: "provided by the whole medical devices"; line 146 "sodium chloride solution in whom.."; etc)

Response 1: This version of the manuscript has been revised by a native speaker.

Point 2: Repeatedly, the authors give mean +/- SD data in the text, while providing median and IQR data in the tables. This makes it challenging to the reader to understand which data the authors are writing about. Please harmonize, I assume data are normally distributed and give mean +/- SD (+n-number) data to more fully inform what the measurements looked like.

Response 2 : Data provided in tables are now expressed in mean ± SD in order to harmonize with data reported in the text.

Point 3 : I believe the order of appearance of the tables has been mixed up. Line 21 to 217: I believe the data in this abstract pertain to table 4 (data on BPA and its derivatives in dialyzer rinsing solution), not to table 2 as is written now.

Response 3: The order of appearance of the tables has been modified.

Point 4: Please provide a concise description of what the ex vivo dialyzer model loooked like. Since dialysate and blood levels were almost equal what did time and flow rates look like? What were BPA levels in the used dialysate to start with?

Response 4: This ex-vivo model was described in our previous publication (Bacle, A. et al. IJP 2016). Dialysate and blood compartments were recirculated countercurrently with 400 mL of BPA-free water at 250 mL/min for 180 min. Experiments were all carried out at 37°C, in order to mimic the conditions of a hemodialysis session. This point was added in the revised version of the manuscript page 6, line 150.

No dialysate was used in these experiments.

Reviewer 2 Report

This is an interesting manuscript well written and performed that deals with novel insights into the BPA field. I think this manuscript provide new routed of BPA exposure not well established yet.

Minor comments: Line 115: the source of BPA-free water is not specified.

Line 158: The authors used different BPA concentration units for calibration (mg.L) and for presenting data (ng.L) a short explanation would be useful for the readers.

Author Response

We sincerely appreciate the comments of the reviewers.

Point1 : Line 115: the source of BPA-free water is not specified.

BPA-free water was purchased from Carlo Erba (Val de Reuil, France). This point was added to the revised manuscript, page 4, line 116.

Point 2: The authors used different BPA concentration units for calibration (mg.L) and for presenting data (ng.L) a short explanation would be useful for the readers.

A short explanation was added to the revised manuscript page 6, line 174.

Reviewer 3 Report

The unexacted or accidental exposure to bisphenol A (BPA) and its chlorinated derivatives to people suffering from end-stage renal disease (ESRD) through online hemodiafiltration (OL-HDF) is of serious concern. Authors clearly dictated various routes of BPA occurrence, consequent exposure to humans and the possible associated risks. Especially considering lack of studies/knowledge available on chlorinated derivatives of BPA, this article certainly finds global interest. The analytical data, statistical relevance and other supporting evidences are acceptable.

However, there are some minor/major comments need to be addressed.

Major comments

Chlorination and chloramination are two major treatment process for drinking water. However, authors did not mention or discussed about derivatives/by-product of BPA resulting from chloramination process.

As authors specified, dialyzers are rinsed with 0.9% sodium chloride before HDF session. Are there any possible effects of sodium chloride in the production of chlorinated derivatives of BPA?

Section 2: Please add purity and purchased details for the standards used. Also, specify and add details about the quality control experiments.

Table 2: Why there is an increase in BPA concentration from SP5 to SP6? Need to be explained in the text.

Line 226-227: I assume that these concentrations are discussed in mean±SD? It is worth to add mean±SD values in the table 2.

Line 307-309: Authors indicated the ultrafilters could be a major source of BPA. Did authors perform any systematic leaching study on these ultrafilters?

It would add great values if authors measured the concentrations of these chemicals in patient’s specimen before and after treatment (OL-HDF)?  

Can authors comments on stability of chlorinated BPA? For example, are they inter-convertible or can produce BPA?

Minor comments

Keywords: Split the key words “End-stage renal diseases bisphenol A”. Keywords should be shorter and unique.

Table 3: Please remove BPA from the table footnote since it is not included in the table.

References: References 30 and 31 are same. Please, delete one of them and re-organize the following references.

Author Response

Thank you very much for your valuable and constructive suggestions. We have carefully revised the manuscript according to your suggestions.

Major comments

Point1 : Chlorination and chloramination are two major treatment process for drinking water. However, authors did not mention or discussed about derivatives/by-product of BPA resulting from chloramination process.

Response 1: This point has been added to the conclusion of the revised manuscript, page 13, line 409.

Point 2: As authors specified, dialyzers are rinsed with 0.9% sodium chloride before HDF session. Are there any possible effects of sodium chloride in the production of chlorinated derivatives of BPA?

Response 2: Because BPA chlorination consists in an electrophilic substitution mechanism, facilitated by alkaline pH and occurring with the Cl+ entity, this reaction  with sodium chloride (Cl-) is unlikely.

Point 3: Section 2: Please add purity and purchased details for the standards used. Also, specify and add details about the quality control experiments.

Response3: Purity and purchased details have been added page 6, line 158. Quality control experiments were already described on page 7, line 202.

Point 4 : Table 2: Why there is an increase in BPA concentration from SP5 to SP6? Need to be explained in the text.

Response 4: BPA might be eliminated during the different treatment steps and in the meantime, continuously released in the system via the different housings and materials. This point has already been discussed (see page 12, line 324).

Point 5: Line 226-227: I assume that these concentrations are discussed in mean±SD? It is worth to add mean±SD values in the table 2.

Response 5: Data provided in tables are now expressed in mean ± SD in order to harmonize with data reported in the text.

Point 6: Line 307-309: Authors indicated the ultrafilters could be a major source of BPA. Did authors perform any systematic leaching study on these ultrafilters?

Response 6: The results of these experiments were already reported on page 9, line 262.

Point 7 : It would add great values if authors measured the concentrations of these chemicals in patient’s specimen before and after treatment (OL-HDF)?  

Response 7: Determination of theses environmental pollutants in urine is considered as the gold standard method due to its non-invasive sampling method and to the large volume available. However, urine collection may be compromised in anuric patients. Therefore, BPA and ClxBPA determination in blood is highly relevant for biomonitoring in these populations. To our knowledge, only one study has described simultaneous quantification of BPA and ClxBPAs in human serum, but the validation of the reported analytical method does not comply with international guidelines. We therefore aimed to develop a fully validated Ultra High Performance Liquid Chromatography coupled with tandem Mass Spectrometry method to determine BPA and its chlorinated derivatives in human blood plasma. This method will be suitable to assess ultratrace levels of BPA and ClxBPAs in human plasma and will be applied to assess plasma levels of these compounds before and after a treatment session.

Point 8 : Can authors comments on stability of chlorinated BPA? For example, are they inter-convertible or can produce BPA?

Response 8: No long term stability study exists for ClxBPA. However, our own experimental unpublished data suggest long-term stability of ClxBPA in water. Therefore, BPA and ClxBPA concentrations found in water are not supposed to be influenced by ClxBPA transformation.

These elements have been added to the revised manuscript, page 12, line 329.

Minor comments

Point 9 : Keywords: Split the key words “End-stage renal diseases bisphenol A”. Keywords should be shorter and unique.

Response 9: There was a mistake in the punctuation. The key words have been modified.

Point 10 : Table 3: Please remove BPA from the table footnote since it is not included in the table.

Response 10 : BPA was removed from the table footnote (see page page 11, line 295)

Point 11 : References: References 30 and 31 are same. Please, delete one of them and re-organize the following references.

Response 11 : The references were re-organized.

Round 2

Reviewer 1 Report

All issues raised were addressed. The manuscript has improved significantly from the revision.

Reviewer 3 Report

All queries were well responded by authors and the corresponding texts/data have been updated in the revised version. I'm pleased with the current form of the manuscript and would recommend for publication in biomolecules. 

This manuscript is a resubmission of an earlier submission. The following is a list of the peer review reports and author responses from that submission.